# Different Maize Ear Rot Fungi Deter the Oviposition of Yellow Peach Moth (*Conogethes punctiferalis* (Guenée)) by Maize Volatile Organic Compounds

Yinhu Chen [1,†], Jie Han [1,†], Haiqing Yang [2], Xiaochun Qin [1], Honggang Guo [1,*] and Yanli Du [1,*]

1    College of Bioscience and Resource Environment/Key Laboratory of Urban Agriculture (North China), Ministry of Agriculture and Rural Affairs of the People's Republic of China, Beijing University of Agriculture, Beijing 102206, China
2    Beijing Pinggu District People's Government Fruit Office, Beijing 101200, China
*    Correspondence: guohonggang1@163.com (H.G.); 20057202@bua.edu.cn (Y.D.)
†    These authors contributed equally to this work.

**Abstract:** Yellow peach moth (*Conogethes punctiferalis* (Guenée), (Lepidoptera: Crambidae), YPM) and maize ear rot are important pests and diseases of maize (*Zea mays* L., (Poales: Poaceae)). In recent years, YPM has become the most destructive maize pest in the Huang-Huai-Hai summer maize region of China via the tunneling of larvae into maize ears. Interestingly, YPM infestation aggravates the occurrence of maize ear rot and causes heavier yield loss of maize in the field. However, few studies report whether maize ear rot would also affect the behavior of YPM. Here, we identified the effects of maize ear rot caused by four different fungi on maize ears' volatile organic compounds (VOCs) and the cascading effects on the behavior of YPM. The current results found that mated YPM females showed a preference for mock-inoculated maize ears (MIM) or mechanically damaged maize ears (MDM) but showed repellence to *Penicillium oxalicum* (Eurotiales: Aspergillaceae)-infected maize ears (POM), *Trichoderma asperellum* (Hypocreales: Hypocreaceae)-infected maize ears (TAM), *Aspergillus phoenicis* (Eurotiales: Aspergillaceae)-infected maize ears (APM), *Aspergillus flavus* (Eurotiales: Aspergillaceae)-infected maize ears (AFM) in the oviposition selection and four-arm olfactometer experiments, indicating that VOCs emitting from fungi-infected maize ears were all repellent to mated YPM females. Further analyses showed that 57 VOCs were identified from all treatments. The partial least squares discriminant analysis (PLS-DA) displayed a separation between TAM, APM, AFM and POM, MDM, and MIM, with 24.3% and 19.1% explanation rates of the first two PLS components. Moreover, the relative quantities of eight common VOCs from different treatments were lower, and the other three common VOCs were higher in fungi-infected maize ears than those in MIM or MDM. There were also 17 unique VOCs in fungi-infected maize ears. In conclusion, these results suggested that maize ear rot negatively affected the behavior of YPM by changing both components and proportions of maize ears' VOCs. These behavior-modifying VOCs may form the basis for the development of attractant or repellent formulations for YPM's management in the future.

**Keywords:** yellow peach moth (YPM); maize ear rot; tripartite interactions; host plant selection behavior; host plant VOCs

## 1. Introduction

In the agroecosystem, host plants are closely associated not only with herbivorous insects but also with fungi. Previous studies have demonstrated that plant-pathogenic microbes, including protists, bacteria, and fungi, can drastically modify plant metabolisms and, therefore, may have important consequences for the diversity and composition of herbivorous insect species [1–8]. Thus, it is meaningful to investigate tripartite interactions among plant-pathogenic microbes, host plants, and herbivorous insects to improve our knowledge on co-evolutionary theories of plant–fungi–insect interactions and design more

effective management strategies for plant-pathogenic fungi and herbivorous insects in the agroecosystem.

Plant-pathogenic microbes have been reported as direct or indirect drivers for host plant-herbivorous insect interactions, but the outcomes of plant-pathogenic microbes on herbivorous insects' behaviors are inconsistent. A few studies demonstrate the attraction of herbivorous insects to plant-pathogenic fungi-infected host plants, while others suggest repellence or no effect [7,8]. For example, yellow peach moth (YPM) females (*Conogethes punctiferalis*(Guenée), (Lepidoptera: Crambidae), YPM) were attracted to and laid more eggs on different *Penicillium* infected apples (*Malus domestica* M., (Rosales: Rosales)) than non-infected apples [9]. However, the bird cherry oat-aphids (*Rhopalosiphum padi*, (Hemiptera: Aphididae)) prefer uninfected over *Neotypphodium*-infected wild grass alpine timothy (*Phleum alpinum* (Poales: Poaceae)). In contrast, the beetles (*Oulema melanopus*, (Coleoptera: Chrysomelidae)) have no preference for *Neotyphodium*-infected and uninfected wild grass [10]. Given the important role of plant-pathogenic fungi, recent studies show that larval western bean cutworm (*Striacosta albicosta*, (Lepidoptera: Noctuidae)) and YPM infestation could significantly aggravate the occurrence of maize ear diseases and cause heavier yield loss of maize (*Zea mays* L., (Poales: Poaceae)) [11,12], which triggers us to explore whether maize ear disease could also affect the behavior of YPM.

Host plant volatile organic compounds (VOCs) are primary olfactory cues employed by herbivorous insects to locate and orient to food sources and oviposition sites [13,14]. Changes in the components and/or proportions of host plant VOCs caused by fungi infection could be perceived by herbivorous insects, which could then result in the behavioral responses of herbivorous insects from attractiveness to repellence or vice versa [7,15,16]. For example, the infection of *Penicillium* fungi alters apple fruits' VOC profiles, including inducing 16 novel VOCs and increasing the absolute contents of ethyl hexanoate and (*Z*, *E*)-*α*-farnesene, which results in strong attractiveness to mated YPM females [9]. Moreover, except for host plant VOCs, VOCs from the fungus itself (*Melampsora larici-populina*, (Pucciniales: Melampsoraceae)) also attract herbivorous insects (*Lymantria dispar*, (Lepidoptera: Lymantriidae)) [5], suggesting that VOCs emitted from host plants and fungi are both olfactory signals for establishing tripartite interactions among herbivorous insects, host plants, and plant-pathogenic fungi [17,18].

The YPM is a generalist herbivorous insect that attacks more than 100 plant species, such as maize, chestnut (*Castanea mollissima* Blume, (Fagales: Fagaceae)), sunflower (*Helianthus annuus* L., (Campanulales: Compositae)), hawthorn (*Crataegus pinnatifida* Bunge, (Rosales: Rosales)), apple, and so on [19,20], and has become the most destructive maize pest in the Huang-Huai-Hai region of China in the last 20 years [21,22]. The YPM is a multivoltine species with three to four generations per year in North China that overwinters as diapausing larvae [23]. The larvae of YPM are typical fruit borers that infest and use maize ears as food sources and habitats to complete their life cycle in the summer maize region of China [21]. The YPM adult is a nocturnal, herbivorous insect, which feeds, oviposits, and develops primarily in buds and fruits of plants [24]. Furthermore, oviposition is an important event in the life cycle of YPM, and adult females select host plants with optimal features to lay their eggs to maximize offspring fitness [25] If oviposition is prevented, the life cycle of YPM is disrupted, and population growth can be decreased. Moreover, recent studies suggest that sex pheromones and plant-derived attractants or repellents are used as allelochemicals to selectively prevent the mating and/or oviposition behavior of YPM in the integrated pest management of YPM [26,27]. Except for YPM, maize is also affected by maize ear rot, which is one of the most prevalent and destructive diseases of maize. The maize ear rot is caused by more than 40 fungal species alone or together, such as *Fusarium* spp., *Aspergillus* spp., *Trichoderma* spp., and so on [28–30]. The incidence of maize ear rot is commonly ranged from 5% to 20% in 22 provinces in China's fields [31]. The susceptible cultivars can be >50% incidence of maize ear rot in conducive climates, resulting in 30% to 40% yield losses [32]. For example, the incidence of *Penicillium* maize ear rot is about 3% to 90% in some areas of Shanxi, Shaanxi, Hebei, and Tianjin in China [33]. It is the 5%

incidence of maize ear rot in Lingshui, Hainan Province, China, in 2016 [34]. Interestingly, YPM infestation significantly aggravates the occurrence of maize ear rot and causes heavier yield loss of maize [12], suggesting the strong effects of YPM on plant-pathogenic fungi (maize ear rot) and triggering us to explore the effect of maize ear rot on the behavior of YPM.

Here, we hypothesized that fungi isolated from maize ears with maize ear rot could affect the behavior of YPM by altering maize ears' VOCs. To test this hypothesis, the following experiments were carried out: (1) the isolation and purification of fungi from maize ears with maize ear rot to identify the plant-pathogenic fungi; (2) the oviposition selection and four-arm olfactometer experiments among mock-inoculated maize ears, mechanically damaged maize ears, single fungal strain-infected maize ears, and the corresponding fungus growing in potato dextrose agar (PDA) medium to determine the effects of plant-pathogenic fungi on host plant selection behavior of mated YPM females; (3) VOCs collection and analysis of the above-mentioned treatments to clarify the effects of plant-pathogenic fungi on maize ear' VOCs.

## 2. Materials and Methods

### 2.1. Insects

A colony of YPM was established and maintained for about 80 generations on maize ears in climate incubators (RTOP-B, Zhejiang Top Instrument Co., Ltd., Hangzhou, China) at $23 \pm 1$ °C, RH $75 \pm 2\%$, 16L/8D photoperiod, and 3500 lux light intensity. Adult moths were provided with 5–8% honey solution after emergence. Apples covered with gauze pieces were provided for the oviposition of YPM in the cage. The YPM colony kept in climate incubators in the lab was mixed with the field-collected YPM on maize ears every year.

### 2.2. Isolation and Purification of Four Fungal Strains

Four fungal strains (SS1 strain, SS2 strain, SS3 strain, and SS4 strain) were isolated and purified from maize ears with maize ear rot using the traditional tissue separation method. The symptomatic-asymptomatic junction tissues were cut into segments (about 0.5 cm × 0.5 cm) under the horizontal-laminar airflow clean bench and were then immersed completely into 2% NaClO (Sangon biotech Co., Ltd., Shanghai, China) for 3 min and then flushed with sterilized water three times. Next, the junction tissues were immersed completely into 75% ethanol (Beijing Minda Technology Co., Ltd., Beijing, China) for 1 min and then flushed with sterilized water three times. Finally, segments were placed onto PDA (Beijing Aoboxing Biotechnology Co., Ltd., Beijing, China) (containing (g/L): potato 200; dextrose 20; agar 18) medium in Petri dishes, which were then placed in a constant temperature incubator (Shanghai Fuma Experiemental Equipment Co., Ltd., Shanghai, China) at 27 °C. After 4 times of repeated purification, a pure culture of the fungal strain was obtained.

### 2.3. Morphological Identification of Four Fungal Strains

The four fungal strains were identified via their micro-morphological and colony morphological characters. After growing on PDA at 27 °C for 7 days, the colony morphology of the four fungal strains was observed. The micro-morphology of the four fungal strains was observed by light microscopy (Motic BA200, Motic China Group Co., Ltd., Xiamen, China).

### 2.4. Molecular Identification of Four Fungal Strains

Fresh hyphae or conidia were picked and immersed into sterile water in a 1.5 mL centrifuge tube. The suspension of hyphae or conidia was then centrifuged at 10,000 r/min for 3 min. The precipitate was collected and drained with filter paper for further analysis.

Total genomic DNA was extracted directly from pure hyphae or conidia using the rapid extraction kit of fungal genomic DNA (Aidlab Biotechnologies Co., Ltd., Beijing, China) following the manufacturer's instructions. The internal transcribed spacer re-

gion (ITS) of the ribosomal DNA (rDNA) was amplified with ITS1/ITS4 primers (ITS1: 5′-TCCGTAGGTGAACCTGCGG-3′; ITS4: 5′-TCCTCCGCTTATTGATATGC-3′) [35]. Amplification reactions were performed in a 50 µL reaction volume containing 25.0 µL 2 × *Taq* PCR Master Mix (Biomed, Beijing, China), 1.7 µL genomic DNA, 1.0 µL of 100 µmol primer (Sangon Biotec, Beijing), and 22.3 µL ddH$_2$O according to the polymerase chain reaction (PCR) program described in Guo et al. [9]. To further identify the four fungal strains, phylogenetic trees based on their ITS sequences were constructed with the neighbor-joining method in Mega 7.0. Clade stabilities were estimated using bootstrap values, which were generated from 1000 replications.

### 2.5. Pathogenicity Test of Four Fungal Strains

The pathogenicity test was performed using a modified protocol for conidial suspensions [36]. Conidial suspensions of the four fungal strains were prepared by adding 10 mL of sterile distilled water onto 7-day-old cultures of PDA medium and swirling gently to remove the conidia. The conidial suspensions were filtered with a muslin cloth to remove the hyphae and transferred into a centrifuge tube. The conidial suspensions were adjusted to the concentrations of SS1 strain-infected maize ears ($6.4 \times 10^7$ conidia/mL), SS2 strain-infected maize ears ($5.6 \times 10^8$ conidia/mL), SS3 strain-infected maize ears ($6.0 \times 10^7$ conidia/mL), and SS4 strain-infected maize ears ($6.0 \times 10^7$ conidia/mL) with a hemocytometer (XB-K-25 type, Yuhuan precision medical instrument factory, Taizhou, China) for further pathogenicity testing.

Maize ears were sterilized with 75% (*v*/*v*) ethanol, rinsed with sterilized distilled water, and then air-dried on sterilized filter paper under the horizontal-laminar airflow clean bench. Maize ears were cut into pieces (about 80 g) and inoculated using wound inoculation methods. Eight strip-type lesions (about 5 cm length × 0.5 cm depth) were scuffed by a sterile knife at different rows of maize kernels and then inoculated with 1 mL conidial suspensions of the same fungal strain or 1 mL sterile distilled water, which served as the control. The maize ears were placed in sterilized plastic boxes (19 cm × 13 cm × 10 cm) containing 50 mL sterile distilled water to maintain humidity and cultured in a constant temperature incubator at 27 °C for 3–9 days. The symptoms of fungi-infected maize ears were recorded. All experiments were carried out under the horizontal-laminar airflow clean bench.

### 2.6. Maize Treatments

The YPM adult preferred to oviposit on fruits of plants, so the maize ears, but not intact plants, were selected for the following experiments. Maize ears were bought from the supermarket of the Beijing University of Agriculture (Beijing, China). It was not easy to perform the experiments with intact maize ears, so each maize ear was cut into 3–4 pieces (about 80 g/piece). The dissected maize ear pieces were firstly sterilized using 75% alcohol for 2 min and washed with sterilized distilled water under the horizontal-laminar airflow clean bench for further experiments, including oviposition selection experiments, four-arm olfactometer experiments, and VOCs collection.

Mock-inoculated maize ears (MIM): A piece of dissected maize ear (about 80 g) without fungi infection and mechanical damage was prepared as MIM. The MIMs were placed in sterilized plastic boxes (25.7 cm × 18.6 cm × 10.2 cm) and cultured 3 d at 27 °C for further experiments.

Mechanically damaged maize ears (MDM): A piece of dissected maize ear (about 80 g) was scuffed with eight strip-type lesions (about 5 cm length × 0.5 cm depth) by a sterile knife at different rows of maize kernels and then injected with 1 mL of sterilized distilled water. The MDMs were kept and cultured as MIM for further experiments.

Fungi-infected maize ears: A piece of dissected maize ear (about 80 g) was treated as MDM, except that 1 mL conidial suspension was injected instead of sterilized distilled water for *Penicillium oxalicum* (Sphaeropsidales: Discellaceae)-infected maize ears (POM, $6.0 \times 10^7$ conidia/mL), *Trichoderma asperellum* (Moniliales: Moniliales)-infected maize ears

(TAM, $5.6 \times 10^8$ conidia/mL), *Aspergillus phoenici* (Sphaeropsidales: Discellaceae)-infected maize ears (APM, $6.0 \times 10^7$ conidia/mL), and *Aspergillus flavus* (Eurotiales: Trichocomaceae)-infected maize ears (AFM, $6.4 \times 10^7$ conidia/mL). The fungi-infected maize ears were kept and cultured as MIM for further experiments.

### 2.7. Oviposition Selection Experiments

The oviposition selection experiment was carried out to determine the effects of four fungal strains on the oviposition behavior of mated YPM females. For *P. oxalicum*, the dissected maize ear pieces with different treatments, including MIM, MDM, POM, and *P. oxalicum* strain growing in PDA medium (PPO), were separately put into two opposite plastic bowls with punched holes. The bowls were covered with wet cheesecloth and were randomly placed at each one of the four inner corners of a wood-frame cage (40 cm $\times$ 35 cm $\times$ 35 cm) with plastic gauze on the sides to allow oviposition from 10 mated YPM females. The cheesecloths were replaced every day, and the position of each treatment in one cage was changed randomly. The experiment was arranged in a randomized complete block design, with four treatments being one treatment factor. Each of the experimental units was measured for 7 days, so the data collection time (days of YPM oviposition) was the second treatment factor. Each cage was a block, and there were 20 blocks. The egg numbers on each sheet were separately counted per day. The oviposition selection rate (dependent variables) was treated as the egg numbers of one treatment per day (independent variables)/the total egg numbers of four treatments (MIM, MDM, POM, and PPO) per day.

For the other three fungal strains, the oviposition selection of mated YPM females was performed (1) among MIM, MDM, TAM, and *T. asperellum* strain growing in PDA medium (PTA), or (2) among MIM, MDM, APM, and *A. phoenici* strain growing in PDA medium (PAP), or (3) among MIM, MDM, AFM, and *A. flavus* strain growing in PDA medium (PAF) were performed as same as *P. oxalicum* strain.

### 2.8. Four-Arm Olfactometer Experiments

A four-arm olfactometer was used to test the behavioral responses of mated YPM females to the odors among MIM, MDM, POM, and PPO treatments according to the methods in Guo et al. [9]. The behavioral response was classified as a choice if the moth passed over 1/3 length of the arm associated with one of the four odors and stayed there for more than 30 s. Conversely, no choice was assigned if the tested moth remained in the common arm for 3 min. Females that did not make a choice within five min were not included in the statistical analysis. As a result, a total of 85 individual moths (*T. asperellum*), 80 individual moths (*A. flavus*), 84 individual moths (*A. phoenici*), and 82 individual moths (*P. oxalicum*) were used to analyze the responsive rate of females. The responsive rate of females in the four-arm olfactometer experiment was defined as the selection rate, which was calculated as the number of females that made a selection for the odor/the total number of females that made a selection for any odors offered simultaneously. For the other three fungal strains, the details were the same as the *P. oxalicum* strain.

### 2.9. VOCs Collection and Analysis

VOCs from different treatments were collected using the dynamic headspace collection method, as described in Guo et al. [9]. Briefly, two pieces of the dissected maize ear (about 80 g/piece) from each treatment or similar size PDA medium inoculated with the same fungal strain were placed into a 48.3 cm $\times$ 59.7 cm Reynolds oven bag (Reynolds Kitchens, Richmond, VA, USA). The Reynolds oven bags were bought from Amazon (https://gs.amazon.cn, access on 8 November 2022). The bag mouth was tightened with a twist tie around a glass tube (6 mm diameter, 10 cm long) filled with 50 mg of Porapark Q adsorbent (80–100 mesh, Waters Corporation of America Inc., Milford, MA, USA). At the bottom of the bag, a small hole was made and connected to a Teflon tube. The humidified and purified air was pushed into the oven bag at a rate of 450 mL/min, and the VOCs

were trapped by 50 mg porapark Q adsorbent (80–100 mesh, Waters Co.). At the same time, an empty bag was prepared to identify what VOCs were in the background. The collection of VOCs lasted for 4 h from 7:30–11:30 A.M. Five replicates of each treatment were performed and collected at the same time. The trapped VOCs were eluted using 350 μL of chromatography-grade *n*-hexane (Fisher Scientific, 99.9%) and stored at −20 °C. For GC-MS analyses, 200 μL of eluent was subsampled, and 1 μL of *n*-nonyl acetate (TCI, 99.9%) was added as an internal standard.

The identification and quantification of maize' VOCs were carried out using a Shimadzu gas chromatograph (GC) coupled to a Shimadzu mass spectrometer (MS) (GCMS-QP2010E, Japan). The GC was equipped with a DB-5MS column (30 m × 0.25 μm × 0.25 μm). Helium was used as the carrier gas with a constant flow of 36.534 cm/s. The injector temperature was 200 °C, the GC-MS transfer line temperature was 280 °C, the source 230 °C, the quadrupole 150 °C, the ionization potential 70 eV, and the scan range was 30–300 $m/z$. Each 1 μL sample was injected by the mode of splitless. Following injection, the column temperature was maintained at 55 °C for 1 min and then programmed at 5 °C/min to 220 °C (held for 6 min). Compounds were identified by comparing mass spectra with the NIST Standard Reference Database (Shimadzu Co., Kyoto, Japan) and quantified by their total ion abundance relative to that of the internal standard.

### *2.10. Statistical Analysis*

The Shapiro–Wilks test and Bartlett's test were used to check the normality of the distribution and the equality (homogeneity) of the variances of the dependent variables, respectively. Data obtained from oviposition selection experiments were analyzed with repeated-measures analysis to determine the effect of different treatments on the oviposition selection rates of mated YPM females. For *P. oxalicum*, the statistical model included the effect of the treatment group (MIM, MDM, POM, and PPO), days of YPM oviposition, and their interactions. The treatment group differences were subjected to a one-way analysis of variance using Duncan's new multiple-range test at $p < 0.05$. Data from behavioral bioassay in four-arm olfactometer experiments were subjected to one-way analysis of variance using Duncan's new multiple range test ($p < 0.05$). The other three fungal strains were also analyzed as the *P. oxalicum* strain. The relative quantities of VOCs were subjected to a one-way analysis of variance using Duncan's new multiple-range test ($p < 0.05$). All these statistics were analyzed with SPSS 22.0 statistical software (International Business Machines Corporation, Chicago, IL, USA). The partial least squares discriminant analysis (PLS-DA) was performed with the R program (version 4.1.2) (R Foundation for Statistical Computing, Vienna, AUS). Graphs were generated in Graphpad Prism 9.0 (Graphpad software, Boston, MA, USA).

## 3. Results

### *3.1. Morphological and Molecular Identification of Maize Ear-Surface Fungi*

Four fungal strains with different morphology were isolated from the surface of maize ears with maize ear rot, called SS1 strain, SS2 strain, SS3 strain, and SS4 strain. The texture of the SS1 strain was mainly dense velvet, and the central part was flocculent and flat. The colony color of the SS1 strain was yellow to green. It was light at the initial stage and slightly dark after aging (Figure 1A). The conidium of the SS1 strain was spherical (Figure 1B). The SS2 strain had concentric annular colonies with dark green on the PDA medium for 3 days. The aerial hypha of the SS2 strain was sparse (Figure 1C). The conidium of the SS2 strain was oval, and the conidium stalk with slight spiny protrusions was slender and curved (Figure 1D). The colony color of SS3 was dark brown-to-black, and its reverse side was colorless to yellow on the PDA medium (Figure 1E). The conidium was a spherical vesicle, and the conidial head color was light tan to black (Figure 1F). The SS4 strain was velutinous with fluffy and white aerial mycelium that eventually turned green on PDA medium after 3 days. The conidiophore was broom-shaped and conidium was colorless, unicellular, and ellipsoid (Figure 1G,H).

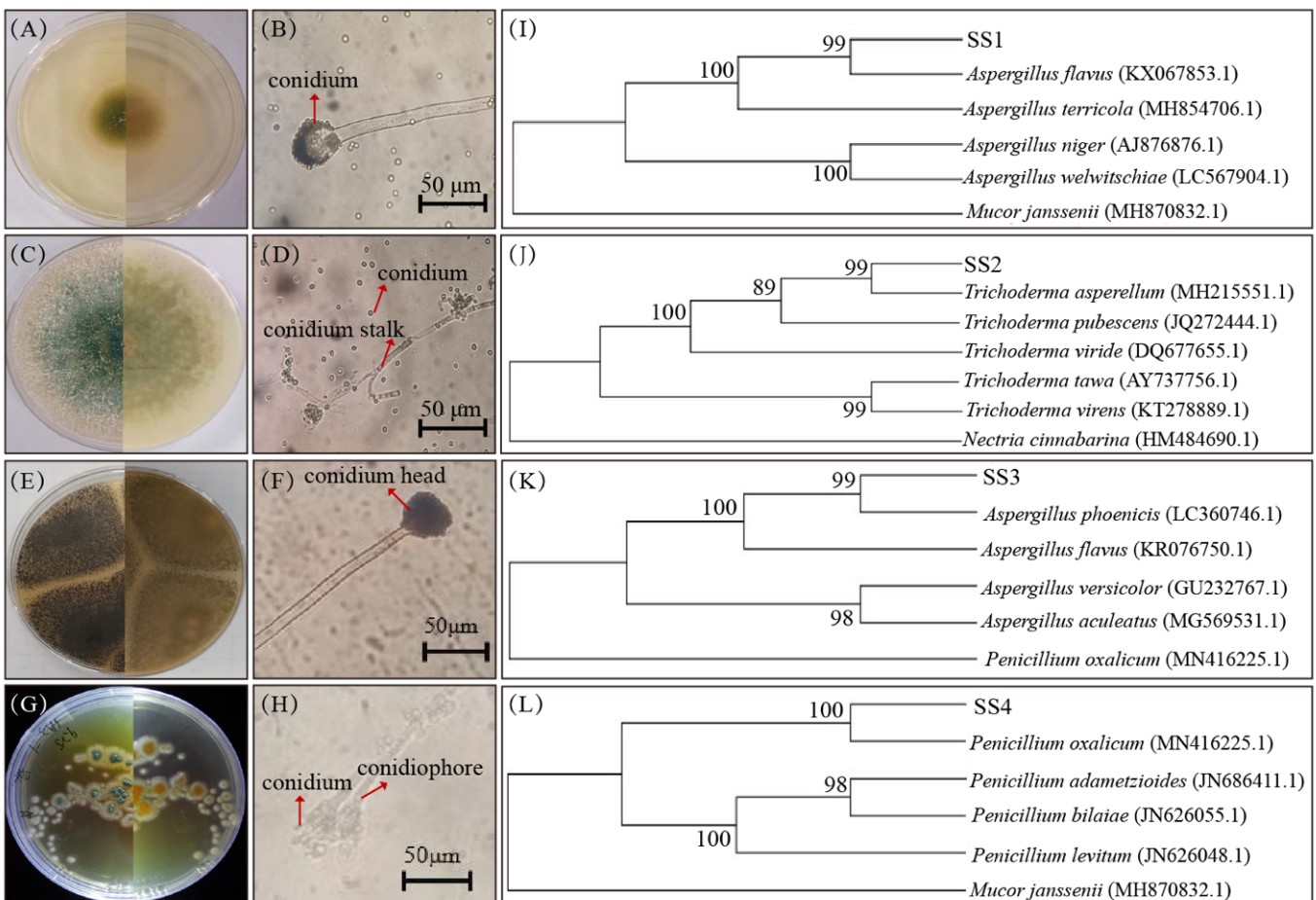

**Figure 1.** Morphological and molecular identification of the SS1 strain, SS2 strain, SS3 strain, and SS4 strain. The morphology of the SS1 strain (**A**,**B**); The morphology of the SS2 strain (**C**,**D**); The morphology of the SS3 strain (**E**,**F**); The morphology of the SS4 strain (**G**,**H**); Phylogenetic trees of four fungal strains: SS1strain (**I**), SS2 strain (**J**), SS3 strain (**K**), and SS4 strain (**L**) with other related strains based on rDNA-ITS sequences by the neighbor-joining method. The scale bar was 50 μm (**B**,**D**,**F**,**H**).

The rDNA-ITS sequences of four fungal strains were cloned for further identification. The SS1 strain (a 566 bp ITS fragment) was clustered with *Aspergillus flavus* ((Eurotiales: Aspergillaceae), KX067853) by a bootstrap value of 99% (Figure 1I). The SS2 strain (a 600-bp ITS fragment) was grouped with *Trichoderma asperellum* ((Hypocreales: Hypocreaceae), MH215551), and the SS3 strain (a 613-bp ITS fragment) was clustered with *Aspergillus phoenicis* ((Eurotiales: Aspergillaceae), LC360746), which was well supported by a bootstrap value of 99% (Figure 1J,K). The ITS sequences of the SS4 strain (a 561-bp ITS fragment) were 100% identical to *Penicillium oxalicum* isolate ((Eurotiales: Aspergillaceae), MN416225) (Figure 1L).

Combined with morphological and molecular characters, the SS1 strain was identified as *A. flavus*, the SS2 strain was identified as *T. asperellum*, the SS3 strain was identified as *A. phoenicis*, and the SS4 strain was identified as *P. oxalicum*.

### 3.2. The Host Preference of Mated YPM Females among MIM, MDM, Fungi-Infected Maize Ear, and the Same Fungal Strain Growing in PDA Medium

For *P. oxalicum* and *A. phoenicis*, the highest oviposition selection rates of mated YPM females were on their respective MDM treatments, followed by those on MIM. The lowest were both on fungi-infected maize ears (POM or APM) and fungi growing in PDA medium (PPO or PAP) along with the extension of infection time from the 1st to 7th day, respectively (Figure 2A,C; Table 1). For *T. asperellum*, the oviposition selection rate of mated YPM females

was the highest on MDM from the beginning till the end (54.43%), which was significantly higher than those on MIM (31.50%), TAM (12.93%), and PTA (1.12%) (Figure 2B; Table 1). As for *A. flavus*, the mated YPM females preferred to lay eggs on MIM during the first three days, then showed preference to MDM in the 4th–5th days, but returned to MIM again on the 7th day. It was worth mentioning that on the first day after infection, the YPM laid much more eggs on AFM than on MDM but turned over from the second day till the end. Considering the average oviposition selection rate of 1–7 d, the oviposition selection rates on MIM (50.20%) and on MDM (39.65%) were both significantly higher than those on AFM (9.64%) and on PAF (0.51%) (Figure 2D; Table 1).

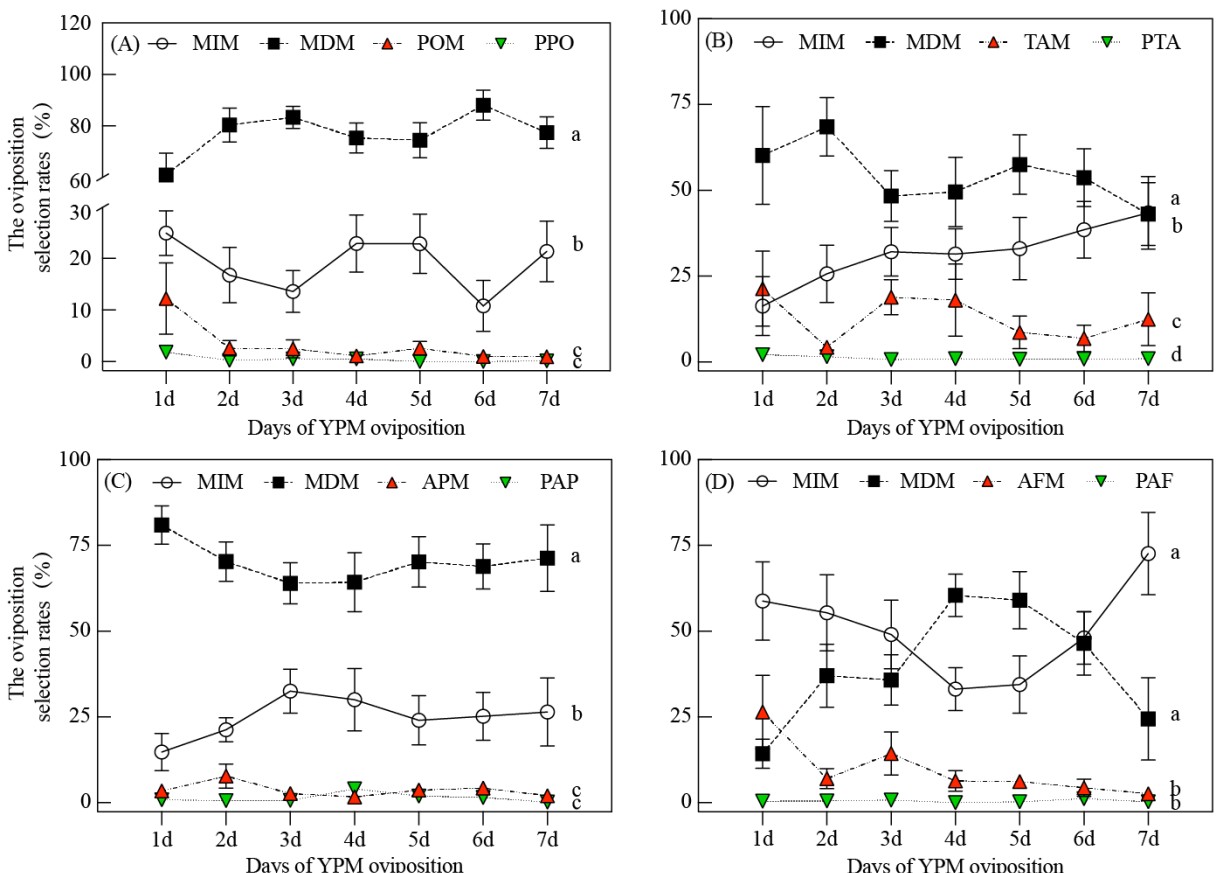

**Figure 2.** The oviposition selection rates of mated YPM females in oviposition behavioral experiments. *P. oxalicum* (**A**), *T. asperellum* (**B**), *A. phoenicis* (**C**), *A. flavus* (**D**). 1–7 d: Days of YPM oviposition; MIM: mock-inoculated maize ears, MDM: mechanically damaged maize ears, POM: *P. oxalicum*-infected maize ears, TAM: *T. asperellum*-infected maize ears, APM: *A. phoenicis*-infected maize ears, AFM: *A. flavus*-infected maize ears, PPO: *P. oxalicum* growing in PDA medium, PTA: *T. asperellum* growing in PDA medium, PAP: *A. phoenicis* growing in PDA medium, and PAF: *A. flavus* growing in PDA medium. Different letters indicate significant differences among the four treatments ($p < 0.05$).

**Table 1.** Effects of fungi infection, days of YPM oviposition, and their interactions on the oviposition selection rates among different treatments by repeated-measures analysis.

| Factor | *P. oxalicum* | | *T. asperellum* | | *A. phoenicis* | | *A. flavus* | |
|---|---|---|---|---|---|---|---|---|
| | *F* Value | *p* Value | *F* Value | *p* Value | *F* Value | *p* Value | *F* Value | *p* Value |
| Days | 0.000 | 1.000 | 0.000 | 1.000 | 0.000 | 1.000 | 0.000 | 1.000 |
| Fungi infection | 120.374 | <0.001 | 48.759 | <0.001 | 84.714 | <0.001 | 3.146 | <0.001 |
| Days × Fungi infection | 2.198 | 0.010 | 1.770 | 0.046 | 0.933 | 0.025 | 2.525 | 0.003 |

A four-arm olfactometer experiment was also prepared to test behavioral responses of mated YPM females to VOCs of different treatments. For *P. oxalicum*, VOCs of MDM showed the most attractiveness to mated YPM females (the selection rate of 41.19%), which was significantly higher than VOCs of MIM (the selection rate of 28.55%), PPO (the selection rate of 18.61%), and POM (the selection rate of 9.01%) ($F_{3,12}$ = 12.337, $p$ = 0.001) (Figure 3A). For *T. asperellum* and *A. flavus*, mated YPM females showed a markedly higher preference for VOCs of MIM and MDM than VOCs of fungi-infected maize ears (TAM or AFM) and fungi growing in PDA medium (PTA or PAF), respectively (Figure 3B,D). For *A. phoenicis*, the selection rate of mated YPM females to VOCs of MIM (36.44%) was significantly higher than that of VOCs of APM (14.72%). The selection rates of mated YPM females to VOCs of MDM (28.70%) and PAP (25.14%) were intermediate ($F_{3,12}$ = 3.040, $p$ = 0.071) (Figure 3C). These results suggested that mated YPM females showed significant repellence to the odor of fungi-infected maize ears (PPO, TAM, APM, and PAF).

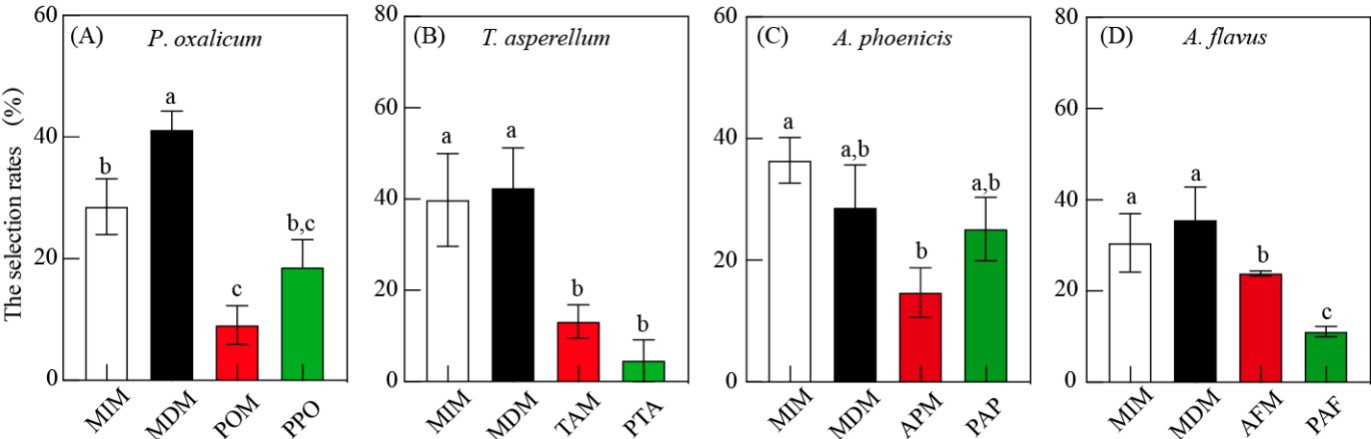

**Figure 3.** The selection rates of mated YPM females in four-arm olfactometer experiments. *P. oxalicum* (**A**), *T. asperellum* (**B**), *A. phoenicis* (**C**), *A. flavus* (**D**). MIM: mock-inoculated maize ears, MDM: mechanically damaged maize ears, POM: *P. oxalicum*-infected maize ears, TAM: *T. asperellum*-infected maize ears, APM: *A. phoenicis*-infected maize ears, AFM: *A. flavus*-infected maize ears, PPO: *P. oxalicum* growing in PDA medium, PTA: *T. asperellum* growing in PDA medium, PAP: *A. phoenicis* growing in PDA medium, and PAF: *A. flavus* growing in PDA medium. Different letters indicate significant differences among the four treatments ($p$ < 0.05).

### 3.3. VOC Profiles of Different Treatments

A total of 57 VOCs were detected in the headspace of different treatments, including 30 VOCs in MIM, 32 VOCs in MDM, 36 VOCs in POM, 24 VOCs in TAM, 34 VOCs in APM, and 24 VOCs in AFM (Table 2). According to the PLS-DA analysis based on the relative contents of all detected VOCs, the first two significant PLS components explained 24.3% and 19.1% of the total variances, respectively. It was worth mentioning that the VOCs of three fungi-infected maize ears (TAM, APM, and AFM) were clearly separated and obviously distinguishable from VOCs of POM, resembling those of MDM and MIM (Figure 4). Furthermore, there were 17 fungi-induced unique VOCs, including (*E*, *E*, *E*)-2,4,6-Octatrien, 2-propylheptanol, methyl benzoate, 2-pentadecanol, 5-propyltridecane in POM, benzaldehyde, 1-octen-3-one, 4-ethylanisole in TAM, 4-methoxystyrene, 3,8-dimethylundecane, 5-butylnonane in APM, 2-ethyl-cyclopentanone and butyl butyrate in AFM, 3-octanone in POM, APM, TAM, 3-octanol in TAM and APM, 3,7-dimethyldecane in TAM, APM, AFM, β-bisabolene in POM, TAM, APM, AFM. Moreover, compared with VOCs in MIM and MDM, the relative quantities of 8 common VOCs, including 2-heptanone, *m*-xylene, 2-heptanol, 3-(1-methylpropyl)- cyclohexene, D-limonene,2-nonanone, ethyl benzoate, and caryophyllene were decreased. The other three common VOCs (1-Octen-3-ol, 1, 3-dichlorobenzene, butylated hydroxytoluene) were increased in fungi-infected maize ears (POM, TAM, APM, AFM) (Table 1).

**Table 2.** VOCs collected from different treatments, including MIM, MDM, POM, TAM, APM, AFM, PPO, PAT, PAP, and PAF [I].

| No. | Compound Type | Retention Time (min) | Compounds [II] | MIM [III] | MDM [III] | POM [III] | TAM [III] | APM [III] | AFM [III] | PPO [III] | PAT [III] | PAP [III] | PAF [III] |
|---|---|---|---|---|---|---|---|---|---|---|---|---|---|
| 1 | Alkane | 8.22 | 3,7-dimethyldecane | 1.0375 ± 0.37 [4] bc | 1.52 ± 0.39 [4] b | 0.9525 ± 0.25 [4] bc | 0.34 ± 0.09 [3] c | | 3.542 ± 0.26 [3] a | | | | |
| 2 | Alkane | 9.427 | Undecane | 45.13 ± 7.46 [4] a | 30.198 ± 3.28 [4] b | 3.42 ± 0.59 [5] c | 0.89 ± 0.14 [5] c | | 1.54 ± 0.37 [4] c | | | | |
| 3 | Alkane | 14.269 | 5-(2-methylpropyl)nonane | 9.23 ± 0.92 [5] b | 11.40 ± 1.91 [5] b | 2.07 ± 0.32 [4] b | 1.82 ± 0.42 [5] b | 1367.61 ± 121.00 [5] a | 3.66 ± 0.70 [3] b | 0.70 ± 0.05 [5] b | 2.19 ± 1.00 [5] b | | |
| 4 | Alkane | 14.289 | 2,3,6,7-tetramethyl-Octane | 11.41 ± 1.81 [5] a | 9.66 ± 0.95 [4] a | 0.65 ± 0.14 [5] b | 0.69 ± 0.12 [4] b | | 1.83 ± 0.39 [5] b | | | | |
| 5 | Alkane | 14.491 | 3,8-Dimethylundecane | 28.85 ± 4.29 [4] a | 32.22 ± 2.12 [4] a | 9.94 ± 2.40 [5] b | 1.19 ± 0.29 [4] c | | | | | | |
| 6 | Alkane | 14.793 | Dodecane | 0.91 ± 0.30 [5] b | 1.05 ± 0.30 [5] b | 22.97 ± 2.44 [5] a | 1.18 ± 0.26 [5] b | | | | | | |
| 7 | Alkane | 14.86 | 5-Butylnonane | 1.74 ± 0.14 [4] bc | 4.12 ± 0.92 [3] b | 3.85 ± 0.09 [4] b | 2.40 ± 0.23 [5] bc | | 2.11 ± 0.72 [5] bc | | 2.31 ± 0.10 [5] bc | 3.33 ± 0.42 [4] b | 11.17 ± 1.84 [3] a |
| 8 | Alkane | 15.494 | 3-methyl-5-propylnonane | | | | | | 10.51 ± 2.15 [5] | | | | |
| 9 | Alkane | 17.372 | 5-Propyltridecane | | | 1.21 ± 0.13 [4] | | | | | | | |
| 10 | Alkane | 17.395 | Tridecane | 1.84 ± 0.25 [5] a | 1.00 ± 0.13 [5] b | | | | | | | | |
| 11 | Alkane | 19.791 | Eicosane | | | | | 17.51 ± 2.82 [5] | | | | | |
| 12 | Alkane | 19.876 | Tetradecane | | | | | 41.42 ± 5.21 [5] | | | | | |
| 13 | Alkane | 20.865 | Nonadecane | | 2.15 ± 0.28 [5] c | 14.08 ± 1.37 [4] c | 277.97 ± 39.74 [5] a | 92.20 ± 20.59 [3] b | 7.11 ± 0.24 [3] c | | 11.21 ± 0.73 [4] b | 82.21 ± 5.23 [5] b | 50.50 ± 12.45 [4] c |
| 14 | Alkane | 22.231 | Pentadecane | | | 2 ± 0.40 [5] b | 12.10 ± 3.92 [5] b | 148.20 ± 9.93 [4] a | | | | | |
| 15 | Alkane | 24.447 | Hexadecane | | | | | | | | | 45.48 ± 1.11 [3] | |
| 16 | Alkane | 24.709 | 2,6,11,15-tetramethyl-Hexadecane | | 14.95 ± 1.08 [5] a | 3.11 ± 0.20 [5] b | 2.63 ± 0.56 [3] b | | | | | | |
| 17 | Alkene | 4.343 | Styrene | | | | 2.54 ± 0.53 [5] b | 15.81 ± 0.57 [3] a | | | | | |
| 18 | Alkene | 5.165 | (2E,4E,6E)-octa-2,4,6-triene | | | | | | 3.69 ± 0.70 [5] | | | | |
| 19 | Alkene | 5.229 | (1R)-(+)-α-piene | 8.79 ± 0.20 [4] d | 13.00 ± 0.44 [4] cd | 21.48 ± 0.78 [5] b | 16.67 ± 2.20 [5] bc | 38.70 ± 4.63 [4] a | 21.82 ± 1.90 [5] b | | | 2.64 ± 0.37 [3] e | |
| 20 | Alkene | 6.52 | 3-butan-2-ylcyclohexene | 4.64 ± 0.14 [4] a | | 2.03 ± 0.19 [4] b | 1.78 ± 0.09 [4] c | | | | | | |
| 21 | Alkene | 7.47 | D-Limonene | | | | | 18.75 ± 2.94 [5] | | | | | |
| 22 | Alkene | 7.493 | 4-methylene-1-(1-methylethyl)bicyclo[3.1.0]hexane | | | | | | | | 1.55 ± 0.15 [5] b | 3.54 ± 0.63 [5] a | |
| 23 | Alkene | 10.788 | 1-ethenyl-4-methoxybenzene | | | | | 0.66 ± 0.15 [4] b | | | 1.63 ± 0.22 [5] b | 3.21 ± 0.49 [5] a | |
| 24 | Alkene | 17.935 | (4Z)-4,11,11-trimethyl-8-methylidenebicyclo[7.2.0]undec-4-ene | | | 1.71 ± 0.25 [4] | | | | | | | |
| 25 | Alkene | 20.127 | β-Bisabolene | | | | 4.72 ± 1.10 [5] a | 1.10 ± 0.30 [4] c | 3.18 ± 0.36 [4] b | | | | |
| 26 | Alcohol | 4.478 | 2-Heptanol | 14.16 ± 1.33 [3] a | 13.93 ± 2.01 [4] a | 0.71 ± 0.11 [3] b | | | | | | | |
| 27 | Alcohol | 6.207 | 1-Octen-3-ol | | | 2.76 ± 0.25 [5] | | | | | | | |
| 28 | Alcohol | 6.607 | 3-Octanol | 2.89 ± 0.67 [5] b | 8.32 ± 1.26 [5] a | | | | | | | | |
| 29 | Alcohol | 8.196 | 2-propylheptanol | 7.75 ± 0.39 [3] c | 7.29 ± 0.63 [5] c | 4.15 ± 0.28 [5] cd | | 16.79 ± 2.61 [5] b | 22.76 ± 3.94 [5] a | | 2.21 ± 0.17 [5] cd | 2.60 ± 0.55 [5] cd | |
| 30 | Alcohol | 9.32 | 2-Pentadecanol | | | 3.00 ± 0.36 [4] | | | | | | | |
| 31 | Alcohol | 9.347 | Linalool | | | | | | | | 1.70 ± 0.27 [5] | | |
| 32 | Alcohol | 10.805 | cis-3-nonenol | | | | 3.63 ± 0.75 [5] | | | | | | |
| 33 | Aldehyde | 5.776 | Benzaldehyde | | | | | 5.48 ± 3.27 [5] | | | | | |
| 34 | Aldehyde | 9.185 | (E)-non-4-enal | | | | | 234.59 ± 102.11 [5] | | | | | |
| 35 | Aldehyde | 9.444 | Nonanal | 1.49 ± 0.53 [5] b | 2.87 ± 0.81 [4] a | | | | | | | | |
| 36 | Aldehyde | 12.232 | Decanal | 3.53 ± 0.05 [4] c | 19.01 ± 3.92 [4] a | 7.01 ± 0.86 [5] bc | 2.49 ± 0.18 [5] c | 12.66 ± 2.66 [5] b | | | 7.06 ± 0.90 [4] bcd | 3.11 ± 0.32 [5] c | 12.29 ± 0.73 [5] b |
| 37 | Ester | 4.052 | Isoamyl acetate | | 4.72 ± 0.99 [4] a | 2.73 ± 0.34 [4] b | 0.87 ± 0.11 [4] c | | | | | | |
| 38 | Ester | 4.967 | Methyl hexanoate | 3.00 ± 0.58 [5] d | 3.41 ± 0.42 [5] cd | 1.73 ± 0.34 [5] d | 1.23 ± 0.18 [5] d | 6.14 ± 1.32 [5] b | 11.95 ± 1.65 [4] a | | 1.53 ± 0.27 [5] d | 1.38 ± 0.13 [4] d | |
| 39 | Ester | 6.636 | Butanoic acid, butyl ester | 3.14 ± 0.40 [4] d | 4.13 ± 0.47 [5] cd | 3.03 ± 0.15 [4] d | 10.30 ± 1.19 [5] b | 10.44 ± 1.42 [4] b | 15.10 ± 1.51 [5] a | | | | |
| 40 | Ester | 9.175 | Benzoic acid, methyl ester | | | | | | | | 1.75 ± 0.27 [5] | | |
| 41 | Ester | 11.275 | Ethyl benzoate | | | | 2.41 ± 0.18 [5] | | | | | | |
| 42 | Ester | 16.075 | Malonic acid, bis(2-trimethylsilylethyl ester | 2.26 ± 0.24 [5] c | 3.41 ± 0.32 [5] b | 1.28 ± 0.16 [4] c | 2.01 ± 0.33 [4] c | 5.11 ± 0.56 [5] a | 4.06 ± 0.46 [5] b | | | | |
| 43 | Ester | 30.474 | Ethyl palmitate | | | | 2.50 ± 0.30 [5] | | | | | | |
| 44 | Ketone | 4.291 | 2-Heptanone | 2.11 ± 0.31 [5] b | 3.29 ± 0.48 [5] bc | 1.60 ± 0.18 [4] c | 7.29 ± 0.77 [5] a | | 7.89 ± 1.09 [5] a | | | | |
| 45 | Ketone | 5.303 | 2-ethyl-Cyclopentanone | 5.01 ± 1.13 [5] cd | 2.78 ± 0.17 [3] d | 6.14 ± 0.95 [5] c | 3.31 ± 0.31 [4] cd | 14.99 ± 0.71 [3] a | | 4.93 ± 0.60 [5] cd | 2.89 ± 0.07 [5] d | 3.76 ± 0.86 [4] c | 9.83 ± 1.04 [4] b |
| 46 | Ketone | 6.192 | 1-Octen-3-one | | | 4.24 ± 0.37 [5] | | | | | | | |
| 47 | Ketone | 6.379 | 3-Octanone | 3.56 ± 0.22 [4] cd | 6.89 ± 0.59 [5] c | | 4.02 ± 0.5 [5] cd | 15.04 ± 1.57 [5] a | 10.29 ± 1.05 [5] b | | 1.73 ± 0.19 [5] d | 2.4 ± 0.58 [5] d | |
| 48 | Ketone | 6.463 | 3-methyl-2-Heptanone | 4.12 ± 0.74 [4] a | 3.79 ± 0.48 [5] a | 0.78 ± 0.01 [3] e | | | | | | | |
| 49 | Ketone | 9.096 | 2-Nonanone | 4.01 ± 0.24 [5] e | 6.40 ± 0.58 [5] d | 4.16 ± 0.75 [5] e | 10.72 ± 0.77 [5] c | 15.13 ± 1.24 [4] a | 13.43 ± 0.41 [5] b | | | | |
| 50 | Aromatic | 4.379 | m-Xylene | 6.69 ± 0.775 c | 10.53 ± 0.775 b | 7.30 ± 0.735 c | 3.97 ± 0.375 de | | 12.77 ± 1.435 a | | 2.10 ± 0.544 e | 2.70 ± 0.484 de | |
| 51 | Aromatic | 4.833 | Anisole | | | 8.32 ± 1.04 [5] b | 15.93 ± 1.81 [5] b | 81.49 ± 16.76 [5] a | 13.83 ± 0.77 [4] b | | | | |

**Table 2.** *Cont.*

| No. | Compound Type | Retention Time (min) | Compounds [II] | MIM [III] | MDM [III] | POM [III] | TAM [III] | APM [III] | AFM [III] | PPO [III] | PAT [III] | PAP [III] | PAF [III] |
|---|---|---|---|---|---|---|---|---|---|---|---|---|---|
| 52 | Aromatic | 7.017 | 1,3-Dichlorobenzene | 3.11 ± 1.20 [5] d | 4.20 ± 1.09 [5] d | 12.55 ± 1.05 [4] b | 6.19 ± 1.62 [3] cd | | 66.86 ± 3.15 [4] a | 2.2 ± 0.62 [5] de | 2.89 ± 0.99 [5] de | 10.86 ± 1.07 [3] bc | 11.95 ± 2.71 [3] bc |
| 53 | Aromatic | 7.984 | m-Diethylbenzene | 3.88 ± 0.15 [5] b | 4.53 ± 0.72 [5] b | 3.61 ± 0.59 [5] b | 7.86 ± 0.40 [5] a | 24.90 ± 1.754 a | 8.89 ± 1.22 [4] a | | | | |
| 54 | Aromatic | 8.143 | 1,4-Diethylbenzene | 7.04 ± 0.915 d | 10.05 ± 0.965 c | 10.90 ± 1.455 bc | 4.07 ± 0.395 de | | 14.83 ± 3.404 b | 2.09 ± 0.505 de | 2.99 ± 0.424 de | | |
| 55 | Aromatic | 9.724 | 1-ethyl-4-Methoxybenzene | 4.54 ± 0.955 cd | 4.83 ± 0.885 cd | 5.23 ± 1.355 cd | 2.52 ± 0.65 d | 5.74 ± 2.765 c | 11.62 ± 4.714 b | 2 ± 0.273 d | | | 12.75 ± 2.205 a |
| 56 | Aromatic | 20.223 | Butylated Hydroxytoluene | 1.53 ± 0.272 b | 5.25 ± 0.484 a | 4.28 ± 0.954 a | 0.55 ± 0.112 b | | | | | | |
| 57 | Others | | 2-Amino-m-cresol, N,O-bis(trimethylsilyl)- | 2.71 ± 0.214 b | 2.56 ± 0.395 b | 2.49 ± 0.145 b | 2.54 ± 0.075 b | 5.00 ± 1.005 a | 4.94 ± 0.514 a | | | | |

[I] MIM: mock-inoculated maize ears; MDM: mechanically damaged maize ears; POM: *P. oxalicum*-infected maize ears; TAM: *T. asperellum*-infected maize ears; APM: *A. phoenicis*-infected maize ears; AFM: *A. flavus*-infected maize ears; PPO: *P. oxalicum* growing in PDA medium; PTA: *T. asperellum* growing in PDA medium; PAP: *A. phoenicis* growing in PDA medium; PAF: *A. flavus* growing in PDA medium. [II] Compounds are confirmed by the comparison of mass spectrum and retention time with the NIST Standard Reference Database (Shimadzu Co., Japan). [III] Relative quantities are quantified by their ion abundance relative to that of the internal standard. Values are mean ± SE ($n$ = 2~5). Values followed by different superscript numbers mean identification times in the same treatment. Different letters within the same row indicate significant differences among treatments ($p < 0.05$).

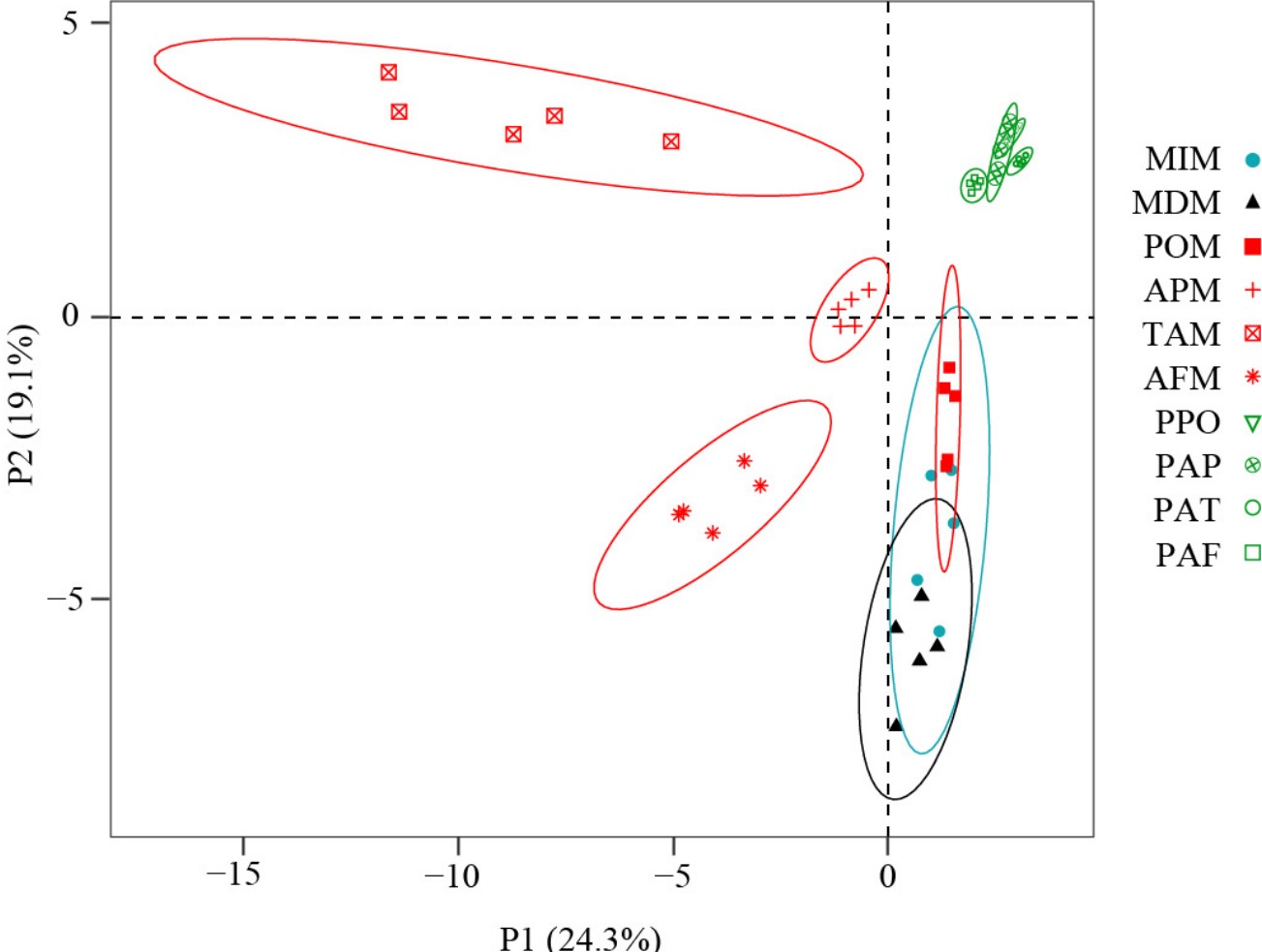

**Figure 4.** Partial least squares discriminant analysis (PLS-DA) of maize ears' VOCs. MIM: mock-inoculated maize ears; MDM: mechanically damaged maize ears; POM: *P. oxalicum*-infected maize ears; TAM: *T. asperellum*-infected maize ears; APM: *A. phoenicis*-infected maize ears; AFM: *A. flavus*-infected maize ears; PPO: *P. oxalicum* growing in PDA medium; PTA: *T. asperellum* growing in PDA medium; PAP: *A. phoenicis* growing in PDA medium; PAF: *A. flavus* growing in PDA medium. P1 and P2: the first two significant PLS components.

## 4. Discussion

Plants are closely associated not only with herbivorous insects but also with fungi, which leads to intimate and complex tripartite interactions. The illustration of tripartite interactions will be beneficial for designing durable ways to integrate the management of plant-pathogenic fungi and pests in agroecosystems. Here, we investigated the effects of maize ear rot caused by *A. flavus*, *T. asperellum*, *A. phoenicis*, and *P. oxalicum* on the behavior of YPM. Mated YPM females were repelled by fungi-infected maize ears (POM, TAM, APM, and AFM) but were attracted to MDM or MIM. Furthermore, compared with MIM and MDM, there were not only 17 fungi-induced unique VOCs but also a decrease in the relative contents of 8 common VOCs and an increase in the other three common VOCs in fungi-infected maize ears, indicating that plant-pathogenic fungi infection could change both components and proportions of host plant VOCs. These findings shed light on the chemical cues of plant-pathogenic fungi in modulating YPM behaviors.

The most common maize ear rot is caused by fungi in the genus *Fusarium*, such as *Fusarium miscanthi* (Tuberculariales: Tuberculariaceae), *Fusarium proliferatum* (Tuberculariales: Tuberculariaceae), and *Fusarium verticillioides* (Tuberculariales: Tuberculariaceae) [28–30]. Except for *Fusarium* spp., more than 40 fungal species, such as *Aspergillus* spp., *Trichoderma*

spp., and so on, have also caused maize ear rot alone or together [37]. Similarly, four fungal strains were isolated from maize ears with maize ear rot and identified as *A. flavus*, *T. asperellum*, *A. phoenicis*, and *P. oxalicum* by morphological and molecular characters, and further confirmed by Koch's postulates experiments in the present study. For *A. flavus*, it is known as one of the common and serious pathogens of maize ear rot [38]. For the other three pathogens, our results and previous documents also confirm that *Trichoderma* spp. and *Penicillium* spp. cause maize ear disease in tropical and subtropical areas of the world [39,40].

It has been reported that maize ear disease affects the preference and performance of herbivorous insects. For example, *A. flavus*, a main pathogen of maize ear rot, has a negative effect on maize earworm (*Helicoverpa zea*, (Lepidoptera: Noctuidae)) with a nearly 4-fold decrease in the infestation rate in CML322 maize [41], which resembles present results that four different fungi-infected maize ears significantly deterred mated YPM females. Interestingly, a recent study has also shown that females of *Diatraea saccharalis* (Lepidoptera: Pyralidae) without *F. verticillioides* prefer fungi-infected sugarcane plants and females with *F. verticillioides* prefer mock plants, suggesting that *F. verticillioides* gains dissemination benefits by modulating the selection behavior of *D. saccharalis* [42]. In fact, maize borers, such as *Sesamia nonagrioides* (Lepidoptera: Noctuidae) and *Ostrinia nubilalis* (Lepidoptera: Pyralidae), serve as vectors of maize ear rot, including *Fusarium* spp. and *Aspergillus* spp., which improves the dissemination of maize ear rot [43]. Furthermore, it has been shown that YPM infestation significantly aggravates the occurrence of maize ear rot and causes heavier yield loss of maize [44]. Therefore, further experiments will be needed to analyze the behavioral responses of YPM with pathogens of maize ear rot between fungi-infected and mock-inoculated maize ears for understanding the interactions between YPM and maize ear diseases.

Plant-pathogenic fungi alter the host plant selection behavior of herbivorous insects via modifying host plant VOCs [45,46]. Maize plants with plant growth-promoting rhizobacteria (PGPR), such as *Bacillus pumilus* (Bacillales: Bacillaceae) INR-7 and *Bacillus mojavensis* (Bacillales: Bacillaceae), emit fewer VOCs and therefore deter the oviposition of European maize borer [47]. A significant decrease in the emission of maize ears' VOCs, including 2-heptanone, *m*-xylene, 2-heptanol, and so on, was also found in fungi-infected maize ears (POM, TAM, APM, AFM) in the present study. However, we also found that the relative quantities of VOCs, such as 1-octen-3-ol and 1, 3-dichlorobenzene, were higher in POM, TAM, APM, and AFM than those in MIM or MDM. This is consistent with a previous report that fungal endophyte (*Neotyphodium lolii*) decreases the emission of dodecane and increases the emission of 2-ethyl-1-hexanol acetate in perennial ryegrass (*Lolium perenne*, (Poales: Poaceae)), which in part, explains the relatively lower selection rate of African black beetles (*Heteronychus arator*, (Coleoptera: Scarabaeidae)) to *N. lolii*-infected plants [48]. Besides changes in the relative quantities of host plant VOCs, the specific VOCs induced by fungi infection can also serve as chemotaxic markers for modulating herbivorous insects' behavior [15,49]. In the present study, some unique VOCs, including 2-propylheptanol, methyl benzoate, 3-octanone, and so on, were detected in fungi-infected maize ears (POM, TAM, APM, AFM), suggesting a potential role of changes in host plant VOCs in the repellency of mated YPM. Moreover, there are numerous examples that VOCs emitted by the fungus could enhance the attraction of lepidopteran insects to host plants [50–52]. Nonetheless, a few cases also suggest that fungal VOCs have an avoidant effect on the behavior of herbivorous insects [17]. VOCs from the fungus were also collected and identified in our study, in which the relative quantities of some VOCs were significantly different between fungi themselves (PPO, PTA, PAP, PAF) and corresponding fungi-infected maize ears (POM, TAM, APM or AFM).

Furthermore, some specific VOCs in the present study have been confirmed as chemotaxonomic markers in regulating herbivorous insects' behaviors. The 2-heptanone, an important green-leaf volatile of host plants, is very attractive to the wasp *Anagrus nilaparvatae* (Hymenoptera: Mymaridae) at a wide range of concentrations from 0 to 500 n mol,

but 2-heptanone and 2-heptanol have a repellent role in brown planthopper (*Nilaparvata lugens*, (Hemiptera: Delphacidae)) [53]. D-limonene of *Citrus* spp. Fruits are increased by *Aonidiella aurantii* (Hemiptera: Diaspididae) infestation and attract parasitoids *Aphytis melinus* (Hymenoptera: Aphelinidae) [54]. Three dominant pest species of the genus *Adelphocoris* (Hemiptera: Miridae), including *A. suturalis*, *A. lineolatus*, and *A. fasciaticollis*, are attracted to the individual compound *m*-xylene in field trials [55]. In contrast, the 1-octen-3-ol produced by *Tricholoma matsutake* (Agaricales: Tricholomataceae) repels *Proisotoma minuta* (Collembola: Insecta) [56]. Interestingly, the current study showed that the relative quantities of D-limonene, 2-heptanone, 2-heptanol, and *m*-xylene were decreased, while the relative quantity of 1-octen-3-ol was increased in fungi-infected maize ears, suggesting that those VOCs might be involved in the repellence of mated YPM females to the fungi-infected maize ears. Except for common VOCs in all treatments, the fungi-induced unique VOCs can also be important. For example, 3-octanol, a fungi-induced unique VOC in TAM and APM, has inhibitory effects on the attraction of spruce bark beetle (*Ips typographus*, (Coleoptera: Scolytidae)) to pheromone [57]. Methyl benzoate, a fungi-induced unique VOC in POM, exhibits repellent effects against *Cimex lectularius* (Hemiptera: Cimicidae) and *Bemisia tabaci* (Hemiptera: Aleyrodidae) [58,59]. Therefore, further experiments are needed to screen candidate active VOCs from the host plant and fungus itself that can be perceived by YPM, which can be considered as putative VOCs as attractants or repellents of YPM.

In summary, our findings demonstrated that fungi-infected maize ears emitted behavior-modifying VOCs that influence the host recognition and selection of YPM. These behavior-modifying VOCs may be used to produce attractant or repellent lures for field management of YPM in the future.

**Author Contributions:** Conceptualization, Y.D. and H.G.; Investigation, Y.C. and J.H.; Formal Analysis, H.G., Y.C. and J.H.; Visualization, H.G.; Writing—Original Draft Preparation, H.G.; Writing—Review & Editing, H.Y., X.Q., Y.D. and H.G.; Funding Acquisition, Y.D and H.G. All authors have read and agreed to the published version of the manuscript.

**Funding:** This study was supported by the science & technology fund of the Beijing Municipal Commission of Education (grant number KZ202210020027), Science Fund for Young Scholars from the Beijing University of Agriculture to H.G. Guo (grant number QNKJ202103), Beijing University of Agriculture Science and Technology innovation Sparkling support plan (grant number, BUA-HHXD2022004), and 2022 Research and Innovation ability promotion Fund for Young Scholars from the Beijing University of Agriculture to H.G. Guo (grant number, QJKC2022001).

**Data Availability Statement:** The data presented in this study are openly available, and the accession number will be applied after the manuscript is accepted.

**Conflicts of Interest:** The authors declare no conflict of interest.

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
