# Peer review of "Different Maize Ear Rot Fungi Deter the Oviposition of Yellow Peach Moth (Conogethes punctiferalis (Guenée)) by Maize Volatile Organic Compounds"

_agronomy, doi:10.3390/agronomy13010251_

Round 1

Reviewer 1 Report

The manuscript “Maize ear rot deters the oviposition of Yellow Peach Moth (Conogethes punctiferalis) via altering maize ears’ VOCs” paper provides new information on the behavioral responses of YPM and VOC analysis on ear rot infection in maize. This is an exciting piece of work. Although the authors did an excellent job on some parts of the manuscripts, many sections of the introduction, methods, results, and discussion need significant improvements. In the following, I will point out specific comments.

Title:

Include volatile organic compounds (VOCs) instead of VOCs

Abstract:

Include order, family, and complete scientific name for Conogethes punctiferalis

Include the family of maize when you first mentioned it.

You need to include why you are looking at these two biotic stresses. In the field, which stage of maize is infested with YPM?

Please change non-infected to mock-inoculated throughout the manuscript

Line 16-19: please revise these sentences for better clarity

Line 21: what is PLS-DA?

Line 23-27: You need to highlight your significant findings with any future directions

 Introduction:

Line 32-36: Not only fungus but also other groups of organisms such as protists (Weeraddana et al., 2020), bacteria (Rivera et al 2017) and fungi.  

See the below examples:

Bacterial phytopathogen infection disrupts belowground plant indirect defense mediated by tritrophic cascade (2017)

Monique J. Rivera, Kirsten S. Pelz-Stelinski, Xavier Martini, Lukasz L. Stelinski

Infection of canola by the root pathogen Plasmodiophora brassicae increases resistance to aboveground herbivory by bertha armyworm, Mamestra configurata Walker (Lepidoptera: Noctuidae) (2020)

Chaminda De Silva Weeraddana, Victor P Manolii, Stephen E Strelkov, A Paulina de la Mata, James J Harynuk, Maya L Evenden

Line 46-50: Please check the formatting in these lines. Also, include the family name of any plant species when you first described it.

Line 55: How severe is this maize year disease? Do you have any estimate of damage to the maize crop? Please provide some examples.

Line 55-57: You are missing a major link in this paper. You need to include the description of YPM and when this pest occurs in the field. First, you need to introduce this or add this part to the YPM description, starting in line 72. This will show us how biologically relevant this study is to maize cropping systems.

Line 72-73: what host plant families are damaged by YPM? Please highlight the significant families with some important crop species here. Also, include some more information on the biology of YPM. How many eggs do YPM lay for how many days? How long YPM live, description of their mating, flying behavior, etc. Do they overwinter? Other stages of YPM such as larvae and pupae. Does YPM have any special behaviors, or whether YPM is nocturnal or diurnal?

Methods:

Line 92: Did you mix the YPM colony with field-collected or wild YPM? If not, please explain why? Which location do you keep the colony?

Line 111: What morphological characters were used to identify the fungi species?

Line 117: Do you know how many conidia or hyphae were collected for the DNA extraction?

 Line 159: Clearly describe whether you used intact plants or maize ears that were dissected for all the experiments. Also, include why you couldn’t use the intact maize for these experiments. Since you are analyzing the VOCs, this part is more important.

Line 196-199: Please clearly explain how you analyzed the data. What are your independent and dependent variables? What is your experiment design etc.?

Line 204: Why did you choose 30 sec as such a short time to consider choices? How many females were excluded? What was the responsive rate of females? Again explain how you conduct the statistical analysis.

Line 214:

Again I am not clear whether you used a dissected maize ear here .  .. Please clearly mention it here

Your control should be maize ear without infection. Why did you choose an empty bag as a control? You can still use an empty bag to see what odors are in the background.

What time of the day did you collect the VOCs?  Did you conduct all five replicates at the same time? Clearly explain this procedure. Did you use Teflon/ PTFE tubes for this? What is your oven bag made of? Where did you buy this?

Hexane and n-nonyl acetate (manufacturer)

Shimadzu gas chromatograph (model)

What is flow in 36.534 cm/s? can you provide this in ml/min

Did you use chemical standards to confirm your VOC identification further?

Statistical analysis:

You need to provide more details on this. Did you choose parametric or non-parametric tests? Did you test the assumptions before running the parametric tests? Did you transform any data to gain normality?

Results:

Your fungi identification and classification are suitable. If you could, please label your fungi structures in the figures . . .

Line 282: I am a little lost on the various acronyms. What is the acronym for Fungi-infected maize ears?

Line 335 and Table 1: Can you classify these VOCs based on the classes of VOCs? So that

We can easily follow this table. Why did you include CAS in the table?

Discussion:

The overall discussion is written ok. I have more suggestions.

Line 1-17: You don’t need to mention your detailed results here. Just provide significant findings.

Line 19-28: This is more suitable for your introduction. You are missing this information in the introduction.

For any insect species you first mention, please use (order: family).

I understand that you mentioned multiple VOCs emitted in different treatments, however, what potential repellents were identified in this study for further studies? If not, are there any VOCs reordered previously as repellents that you also found in your study? Please provide some clues for the readers. Include this in a separate paragraph. Do you have any examples used as repellent lures in the field? Please include all this information in that paragraph.  

References:

I don’t see many issues in the references. Please double-check your references. 

Author Response

Dear Editor:

Thank you for giving us the opportunity to revise our manuscript (Manuscript ID agronomy-2125220) to your journal. Also, we would like to thank the reviewer 1 for his/her comments and constructive suggestions. These comments and suggestions are valuable and helpful, and the manuscript has been correspondingly revised. According to the details of the revision list, the manuscript has been carefully checked to ensure its meet to the requirements of the journal. Please see the attachment. Hopefully, our revisions to the manuscript can explain the major concerns of the reviewers and the revised manuscript is suitable for re-considering publication in your esteemed journal. Responses to reviewers’ comments are listed as blow. Thanks for your time and dedication. 

Best regards,

Honggang Guo

Reviewer 2 Report

Title: Maize ear rot deters the oviposition of Yellow Peach Moth (Conogethes punctiferalis) via altering maize ears’  VOCs

The authors have addressed a complex tripartite association between fungi (ear rot) and herbivorous insect (Yellow Peach Moth) and the host plant (Maize). Fungal rot species modulate the volatile organic compounds of maize ear to provide chemical cues affecting the behavior of YPMs. In this paper, the authors observed the difference in YPM behavior on plants affected by fungal pathogens like Aspergillus phoenia and Aspergillus flavus (termed APM and AFM here), Trichoderma asperellum (TAM), Penicillium oxalicum (POM), compared to non-infected (NIM) and mechanically damaged (MDM) maize ears and found out that YPMs selected NIM and MDM for laying eggs but not the fungal affected maize ears owing to the change in volatile compounds composition and proportion of the plant.

Overall, the studies were well-planned, and experiments were executed well but there are slight changes that I would like the author to address.

1.      Overall grammar should be re-checked. I pointed out a few here.

2.      Title needs to revise, particularly don’t use abbreviations in the title, such as VOCs

3.      Please add an introductory sentence to the abstract identifying potential gaps and the need to do the current study.

4.      Line 14; why YPMs is used in all MS. Isn’t it single pest species or a complex species? Better to use the singular, such as YPM, rather than YPMs.

5.      Line 22; what do you mean about TAM, APM, AFM and POM, MDM, NIM? All terminology should not be abbreviated if it appears the first time in the text.

6.      Line 3: The title would be more precise if “Via” is replaced by “by”.

7.      Little briefing of Oviposition and its significance should be included in the introduction part.

8.      Line 44, 45: “A few literatures” could be just “literature” or “a few studies”.

9.      Line 47, 48, 49: The sentence “mated yellow peach moth (YPM) females (Conogethes punctiferalis) are attracted to and laid more eggs on different Penicillium fungi-infected apples than on fungi non-infected apples” should be rewritten with correct grammar.

*mated YPM females could be just YPM females

*Are to were,

*plant is either fungal infected or non-infected, so Penicillium affected apples than non-infected apples would be fine.

10.  NIM (wherever used) is a non-infected maize ear; fungi non-infected maize ear seems improper repetition. 

11.  Lane 50: Grammatical error: “to” is unnecessary.

12.  Lane 292-295: A suitable justification for the result “As for A. flavus, the mated YPM females preferred to lay eggs on NIM during the first three days, then showed preference to MDM in the 4th-5th days but returned to NIM again in the 7th day”, would be interesting to know. Is odor repellence the only justification here? Is it solely the VOC from plants or fungus, or common VOCs or fungal-induced VOC causing the effect?

13.  There is no image evidence for oviposition/infestation of insects. It would be a nicer addition to the paper.  

The paper should be considered for publication after addressing the minor revision. 

Author Response

Dear Editor:

Thank you for giving us the opportunity to revise our manuscript (Manuscript ID agronomy-2125220) to your journal. Also, we would like to thank the reviewer 2 for his/her comments and constructive suggestions. These comments and suggestions are valuable and helpful, and the manuscript has been correspondingly revised. According to the details of the revision list, the manuscript has been carefully checked to ensure its meet to the requirements of the journal. Please see the attachment. Hopefully, our revisions to the manuscript can explain the major concerns of the reviewers and the revised manuscript is suitable for re-considering publication in your esteemed journal. Responses to reviewers’ comments are listed as blow. Thanks for your time and dedication.

Best regards,

Honggang Guo

Reviewer 3 Report

An excellent paper on interactions between maze insect pests and diseases. The main issue is the English. See attached document. Note that I corrected the worst grammar errors, but doing a complete English edit for a pdf document is too time consuming. Also, I found all of the acronyms (except VOCs, which is common) to be difficult to keep track of. I would suggest keeping the ones for the fungal species and spelling out the other treatments. 

Author Response

Dear Editor:

Thank you for giving us the opportunity to revise our manuscript (Manuscript ID agronomy-2125220) to your journal. Also, we would like to thank the reviewer 3 for his/her comments and constructive suggestions. These comments and suggestions are valuable and helpful, and the manuscript has been correspondingly revised. According to the details of the revision list, the manuscript has been carefully checked to ensure its meet to the requirements of the journal. Please see the attachment. Hopefully, our revisions to the manuscript can explain the major concerns of the reviewers and the revised manuscript is suitable for re-considering publication in your esteemed journal. Responses to reviewers’ comments are listed as blow. Thanks for your time and dedication.

Best regards,

Honggang Guo

Round 2

Reviewer 1 Report

The manuscript “Maize ear rot deters the oviposition of Yellow Peach Moth (Conogethes punctiferalis) via altering maize ears’ VOCs” paper provides new information on the behavioural responses of YPM and VOC analysis on ear rot infection in maize.

I understand that if you made extensive changes in the revised manuscript, it is also hard to see with lots of tracked changes. However, I encourage the author to include tracked changes in the revised manuscript. Also, please mention what you have changed in the revised manuscript according to the reviewer's comments in the response letter. So that reviewers can see what you have changed in your revised manuscript. I had a tough time finding your revisions with reviewer comments.

Further, I couldn’t find any deleted text in the manuscript, as I suggested. You may have included these deletions. Unfortunately, deletions were not tracked. So, it is tough to see whether you have addressed these reviewer comments entirely. I know that you only highlighted the additions to the manuscript text.

 Anyway, I believe the manuscript is in better shape now. You have answered my questions accordingly. I have a few more changes.

 This title does not read nicely “Maize ear rot has the repellent effect on the oviposition of Yellow Peach Moth (Conogethes punctiferalis) by altering maize ears’ volatile organic compounds”.

 Please change something like this:

Different maize ear rot fungi deter oviposition of the yellow peach moth (Conogethes punctiferalis) by maize volatile organic compounds.

Include order and family within brackets. Please change this throughout the manuscript. I see many places where you didn’t include order and family within brackets.

Yellow peach moth (Conogethes punctiferalis (Guenée), (Lepidoptera: Crambidae).

(Castanea mollissima Blume (Fagales: Fagaceae)

 Table 2: You can classify some of “other” into Aromatic VOCs

Author Response

We would like to thank the reviewer 1 for his/her comments and constructive suggestions. I am sorry for missing the deleted text in the last revised manuscript. And we will include tracked changes in the current revised manuscript. Thank you for giving us the opportunity to revise our manuscript (Manuscript ID agronomy-2125220). Hopefully, our revisions to the manuscript can explain the major concerns of the reviewers and the revised manuscript is suitable for re-considering publication. Please see the attachment. Thanks for your time and dedication.
